# Model-based analysis of sample index hopping reveals its widespread artifacts in multiplexed single-cell RNA-sequencing

Rick Farouni [1,2 ✉], Haig Djambazian [1,2], Lorenzo E. Ferri[3], Jiannis Ragoussis [1,2] & Hamed S. Najafabadi [1,2 ✉]

Index hopping is the main cause of incorrect sample assignment of sequencing reads in multiplexed pooled libraries. We introduce a statistical model for estimating the sample index-hopping rate in multiplexed droplet-based single-cell RNA-seq data and for probabilistic inference of the true sample of origin of hopped reads. We analyze several datasets and estimate the sample index hopping probability to range between 0.003–0.009, a small number that counter-intuitively gives rise to a large fraction of phantom molecules — the fraction of phantom molecules exceeds 8% in more than 25% of samples and reaches as high as 85% in low-complexity samples. Phantom molecules lead to widespread complications in downstream analyses, including transcriptome mixing across cells, emergence of phantom copies of cells from other samples, and misclassification of empty droplets as cells. We demonstrate that our approach can correct for these artifacts by accurately purging the majority of phantom molecules from the data.

[1] Department of Human Genetics, McGill University, Montreal, QC  H3A 0C7, Canada. [2] McGill University Genome Centre, Montreal, QC H3A 0G1, Canada. [3] Department of Surgery, McGill University, Montreal, QC H3G 1A4, Canada. ✉email: tarek.farouni@mcgill.ca; hamed.najafabadi@mcgill.ca

Due to the increasing capacity of modern sequencing platforms, sample multiplexing, the pooling of barcoded DNA from multiple samples in the same lane of a high-throughput sequencer, is rapidly becoming the default option in single-cell RNA-seq (scRNA-seq) experiments. However, as several studies have recently shown[1,2], multiplexing leads to incorrect sample assignment of a significant fraction of demultiplexed sequencing reads. Out of several mechanisms that can introduce sample index missassignment[3], the presence of free-floating indexing primers that attach to the pooled cDNA fragments just before the exclusion amplification step in patterned sequencing flowcells has been shown to be the main culprit[4]. This phenomenon is known as sample index hopping and results in a data cross-contamination artifact that takes the form of phantom molecules, molecules that exist in the data only by virtue of read misassignment (Fig. 1a). See Supplementary Note 1 for a more detailed overview of the phenomenon of sample index hopping.

The presence of phantom molecules in droplet-based scRNA-seq data should be a cause of great concern: they can introduce phantom cells (i.e., cells that do not exist in one sample, but appear only as a result of hopping of reads from cells of another sample); they can cause transcriptomes to mix[2,4] (when hopped reads happen to carry the same cell barcode as one of the cells in the target sample); or they can lead to incorrect classification of barcodes that correspond to empty droplets. As a result, identification of cell subpopulations (especially rare cell types) as well as genes that are differentially expressed across cell types can be confounded in downstream analyses. Importantly, it is conceivable that even when the index-hopping rate is very small, the fraction of phantom molecules can still be high due to the distributional properties of sequencing reads across samples: PCR amplification during library preparation can create several copies of the same molecule, and sample misassignment of the sequencing read from only one of these copies is enough to create one phantom molecule (Fig. 1b).

Despite recent attempts to computationally estimate the rate of sample index hopping in plate-based scRNA-seq data[5,6], no statistical model of index hopping for droplet-based scRNA-seq data has yet been proposed. Consequently, current computational methods can neither accurately estimate the underlying rate of index hopping nor adequately remove the resulting phantom molecules in droplet-based scRNA-seq data. This has been a challenging problem since droplet-based libraries are tagged with a single sample index rather than a unique combinatorial pair of sample indices such as those used in plate-based approaches (see Supplementary Note 2 for more details). As a solution to this problem, we here propose a statistical framework that provides (1) a generative probabilistic model that formalizes in a mathematically rigorous manner the phenomenon of index hopping, (2) a statistical approach for inferring the sample index-hopping rate (SIHR) in droplet-based scRNA-seq data at the level of individual reads, (3) a non-heuristic, model-based approach for probabilistic inference of the true sample of origin of hopped sequencing reads, and (4) a data decontamination procedure for purging phantom molecules that optimally minimizes the false positive (FP) rate of molecule reassignments. We validate our

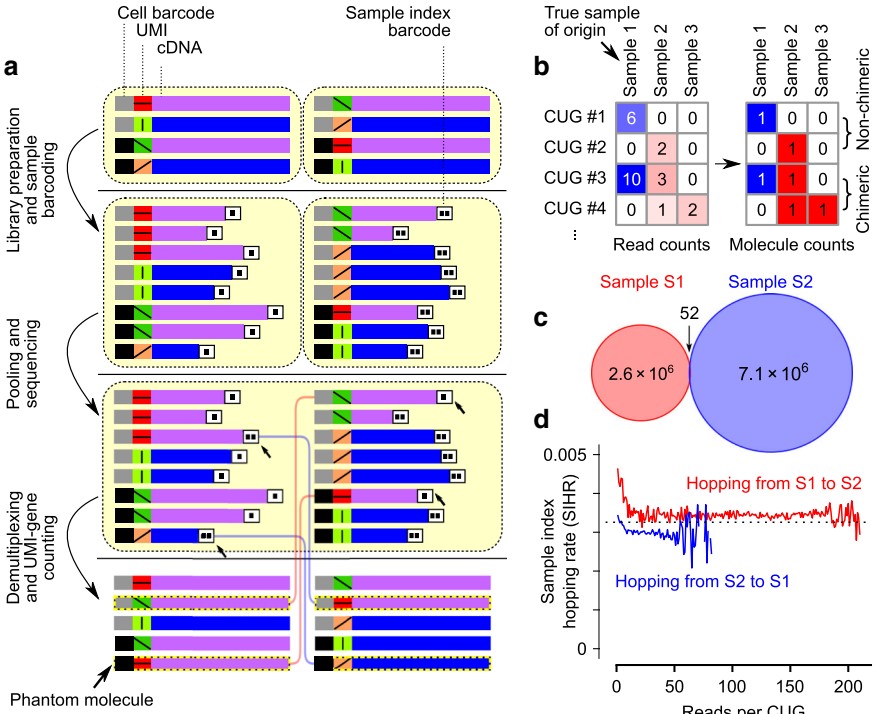

**Fig. 1 A model for sample index hopping. a** A schematic representation of sample index hopping. **b** A toy example showing a read count matrix (left) and the resulting molecule count matrix (right). In this example, the true sample of origin of all reads is sample 1. Blue depicts the reads/molecules that are correctly assigned to their true sample of origin, whereas red represents the hopped reads and the resulting phantom molecules, with the color intensity showing relative counts. A CUG is a unique cell–UMI–gene combination—CUGs that have a nonzero count in only one sample are called non-chimeric, whereas CUGs with nonzero counts in more than one sample are chimeric. See also Supplementary Tables 9–10. **c** Venn diagram showing the number of CUG collisions between two libraries that were sequenced on two separate HiSeq 4000 lanes (see "Methods"). Note the minuscule fraction of CUGs that collide (<6 in a million). **d** The proportion of hopped sample indices by source sample across a range of PCR amplification levels $r$, in a multiplexed dataset with known ground truth (the sample of origin of each CUG was determined based on non-multiplexed sequencing runs of the same libraries on two separate lanes, i.e., **c**). The straight dashed line shows the marginal mean of proportion of hopped reads. Note that the different lengths of the red and blue lines reflect the range of $r$ for which there were enough observations in each sample to calculate the index-hopping rate.

model-based approach using simulated data as well as a multiplexed scRNA-seq dataset in which the true sample of origin of each sequencing read is known. Furthermore, we systematically analyze several previously published multiplexed droplet-based scRNA-seq datasets and show that a substantial fraction of molecules (and cells) in these datasets are artifacts of sample index hopping, but which can be effectively detected and removed using our method.

## Results

**A generative model of index hopping**. The probabilistic model we propose starts with the observation that each cDNA fragment, in addition to its sample barcode index, has a cell barcode and a unique molecular identifier (UMI), and maps to a specific gene (Supplementary Methods). We make two simplifying assumptions to derive a tractable probability distribution for the sample–cell–UMI–gene combinations encoded in the sequencing reads, while taking into account the index hopping. First, as it has been suggested previously[6], we make the assumption that any unique cell–UMI–gene combination (hereafter referred to as CUG) is so unlikely that it cannot arise independently in any two different samples. Accordingly, each CUG would represent one unique molecule and all sequencing reads with the same combination would correspond to PCR amplification products of that original molecule. Second, we assume that the probability of index hopping is the same for all reads, regardless of the source or target sample of the read or the number of other reads that have the same CUG (Supplementary Methods).

To validate the first assumption, we sequenced two 10x Genomics scRNA-seq samples separately on two different lanes of HiSeq 4000, and examined the CUGs that were shared between the sequencing reads of these two samples. As predicted by assumption I, only a minuscule fraction of CUGs from the two libraries were common ($<6$ in every $1 \times 10^6$, Fig. 1c). To validate the second assumption, we then multiplexed the same two libraries on the same lane of HiSeq 4000 and cross-referenced each sequencing read to the non-multiplexed set of sequences, based on the CUG label of the read. This allowed us to infer the true sample of origin of each read and compare it with the apparent sample of origin in the multiplexed experiment. As predicted by assumption II, the rates of sample misassignment were highly similar for reads that originated from any of the two samples (within 10% of the marginal mean of 0.00326), and were largely constant across PCR amplification levels (Fig. 1d).

These two simplifying assumptions allowed us to derive a mixture-of-multinomials model for the observed sample indices of the sequencing reads (see "Methods" and Supplementary Methods). In this generative model, the (unobserved) true sample of origin of each CUG is drawn from a categorical distribution, and the observed sample indices of the reads that originate from that CUG are drawn from a multinomial distribution governed by a single index-hopping parameter (Supplementary Fig. 1).

**Estimating the sample index-hopping rate**. We used this model to further derive a closed-form expression of the probability distribution of chimeric observations, namely, CUGs whose corresponding reads are assigned to multiple samples (Fig. 2a and Supplementary Methods). We were then able to estimate the index-hopping rate by fitting the resulting generalized linear model to the empirically observed distribution of chimeric CUGs across PCR amplification levels (see "Methods" and Supplementary Methods).

Two lines of evidence suggest that this approach provides accurate estimates of SIHR: first, we applied our model to the two-sample multiplexed library described in the previous section

to estimate the SIHR, and then compared it with the empirical SIHR that was calculated based on the ground truth (i.e., based on the true sample of origin of reads). In this dataset, the model-based estimated SIHR was within 5% of the empirical SIHR (Fig. 2b). Second, we observed close agreement between the observed and fitted non-chimeric CUGs across multiple datasets, including two eight-sample HiSeq datasets from mouse epithelial cells[6,7] and two 16-sample NovaSeq 6000 datasets from Tabula Muris[8] (Fig. 2c and Supplementary Fig. 2). Overall, we found that SIHR ranges between 0.3% and 0.9% in the data we analyzed (Supplementary Table 1).

**Model-based purging of phantom molecules**. We used our probabilistic model to also derive a closed-form expression for the posterior distribution of the true sample of origin of each CUG, given its read counts across samples, the SIHR, and the molecular proportions of the samples conditional on the PCR amplification level (see "Methods" and Supplementary Methods). In other words, for each CUG (molecule), we can calculate the probability of belonging to each sample based on its read count distribution across samples (Fig. 3a). We can then assign each CUG to the most likely sample of origin, i.e. the sample with the largest posterior probability, and remove the other phantom molecules, i.e. copies of that CUG that are mistakenly assigned to other samples. Application of this approach to our two-sample multiplexed dataset with the known ground truth revealed that by simply assigning each CUG to the most likely sample, we can remove >97% of the phantom molecules, while keeping >99.9% of the true molecules.

We can further decontaminate the data by removing assignments that have low posterior probability. We have devised an approach for estimating the number of FP and false negatives (FN) for distinguishing true molecules from phantom molecules after decontamination at different posterior probability cutoffs (see Supplementary Methods; Supplementary Fig. 3; Supplementary Table 2). Application of this approach to the multiplexed dataset with known ground truth revealed that the estimated FP and FN counts closely follow the empirical estimates (Fig. 3b). While the FP vs. FN counts relationship can be directly examined to select an appropriate cutoff, our approach also enables the selection of a cutoff based on a user-specified marginal trade-off ratio (TOR), representing the number of real molecules one is willing to discard in order to correctly purge one extra phantom molecule (Supplementary Fig. 4 and Supplementary Methods). We evaluated the TOR cutoff approach by contrasting it with two alternative actions: no purging, where we leave the data as it is, and no discarding, where we assign each CUG to the most likely sample (and therefore purge predicted phantom copies of those CUGs) but refrain from further discarding the CUGs whose inferred sample of origins have low posterior probabilities (equivalent to a TOR cutoff of zero). In most cases, a TOR cutoff of 3 provides a suitable trade-off between FP and FN counts (Fig. 3b, c). Overall, we observed that we can achieve a sensitivity of 0.999 (down from 1 in the original non-purged data) and specificity >0.97 (up from 0 in the non-purged data) in distinguishing true molecules from phantom molecules across all samples (Fig. 3 and Supplementary Tables 3–4).

**Comparison with existing phantom purging approaches**. Using the multiplexed scRNA-seq data with known ground truth, we found that our proposed model-based purging of phantom molecules substantially outperforms a previous heuristic approach[6] that is based on retaining molecules with a certain minimum read fraction (MRF) assigned to one sample (Fig. 3b). For example, at comparable specificity to the default MRF ≥ 0.8

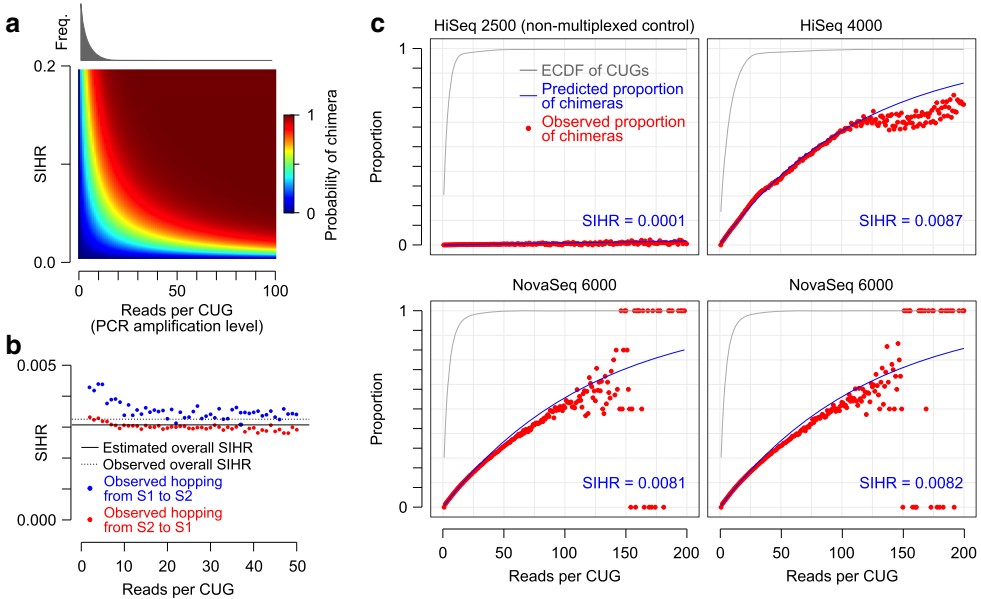

**Fig. 2 Estimation of sample index-hopping rate. a** A heatmap depicting the probability of observing a chimera, i.e., a CUG whose reads map to more than one sample, as a function of sample index-hopping rate (SIHR) and PCR amplification level $r$. See "Methods" for the probability function. The top histogram shows the distribution of $r$ for an example multiplexed HiSeq 4000 dataset[6]. **b** Validation data showing the ground truth proportion of hopped reads by sample conditional on the PCR duplication level alongside the ground truth marginal mean proportion of hopped reads and the model-estimated sample index-hopping rate (same dataset as Fig. 1c, d). **c** Concordance between model predictions and empirically observed number of chimeras for four previously published datasets. Each plot shows the proportion (probability) of chimeras conditional on PCR amplification level, with red dots depicting observed values and blue line representing the model prediction. The ECDF (empirical cumulative distribution function) of $r$ (gray) shows the cumulative proportion of CUGs with at most $r$ reads. Top left: mouse epithelial non-multiplexed HiSeq 2500 dataset[7]; top right: mouse epithelial multiplexed HiSeq 4000 dataset[6]. Bottom left: Tabula Muris NovaSeq 6000 dataset (Lane 1)[8]. Bottom right: Tabula Muris NovaSeq 6000 dataset (Lane 2)[8].

cutoff, our probability-based approach results in almost half as many FN (i.e., true molecules that are incorrectly labeled as phantom molecules; Supplementary Table 3 and Fig. 3b).

To more comprehensively evaluate the performance of our approach, we set out to simulate data in a way that they capture a range of SIHR values as well as different library complexities. In a multiplexed experiment, often samples that vary in their library complexity are sequenced together, where we define library complexity as the expected number of unique molecules sampled with a finite number of sequencing reads generated in a given high-throughput sequencing run. Samples may differ in their total number of unique transcripts due to a host of factors, ranging from the presence of varying amounts of RNA that characterize different cell types to accidental errors in library preparation that could cause cells to break up and lose their endogenous mRNA. In other words, even if the total number of available sequencing reads were budgeted evenly over the multiplexed samples, the number of unique molecules detected in the sequencing run could vary widely across the samples. We formulate the library complexity of a set of multiplexed samples as the molecular proportions of the samples conditional on the PCR amplification level, which we term the molecular proportions complexity (MPC) profile. The MPC profiles of four empirical datasets that we analyzed are shown in Supplementary Fig. 5 (see Supplementary Methods for a more detailed discussion of MPC profiles).

We first used the MPC profile of a library of eight mouse epithelial samples multiplexed on HiSeq 4000[6] (Supplementary Fig. 5b) as the basis for simulating datasets with varying SIHR values. As Supplementary Table 4 and Fig. 3c show, our probabilistic approach gives an FN proportion that is about three times lower than the MRF approach at a comparable FP proportion. More importantly, the entire range of the FP/FN trade-off curve provides a more optimal choice than the MRF

approach, irrespective of the SIHR (Fig. 3c). Furthermore, changing the MRF threshold had little effect on the FP rate, indicating that FP cannot be controlled with this heuristic method. We also simulated additional datasets with two different MPC profiles, and again observed that our probability-based model substantially outperforms the MRF approach across a range of FP/FN TOR (Supplementary Fig. 6), irrespective of the MPC profile.

**Prevalence of phantom molecules in scRNA-seq data.** We applied our model-based approach to quantify and purge phantom molecules across several previously published datasets[6–8], and found that the proportion of phantom molecules (PPM) for individual samples can depart drastically from the marginal PPM of the entire dataset, with sample variation depending on the MPC profile of the multiplexed samples (Fig. 4a). For example, in a library of eight mouse epithelial samples multiplexed on HiSeq 4000[6], two low-complexity samples (i.e., samples in which a relatively small number of unique molecules contribute to the majority of sequencing reads) obtain PPM values exceeding 80% (Fig. 4b and Supplementary Table 5). In other words, in these samples, >80% of the UMI counts correspond to phantom molecules that originated from other samples. The unexpectedly large PPM values can also be observed in other datasets. For example, in the Tabula Muris data[8], PPM of some samples reaches values as high as 17% (Supplementary Table 6). The high abundance of phantom molecules in these samples is surprising given the relatively low index-hopping rate in these datasets (SIHR ≈ 0.008; Supplementary Table 1). As shown in Fig. 4b for the HiSeq 4000 dataset[6], high PPM values seem to largely correspond to samples with high read-to-molecule ratios (RMR), a good proxy measure of the extent of sample-specific PCR amplification bias which we define as the ratio of the total

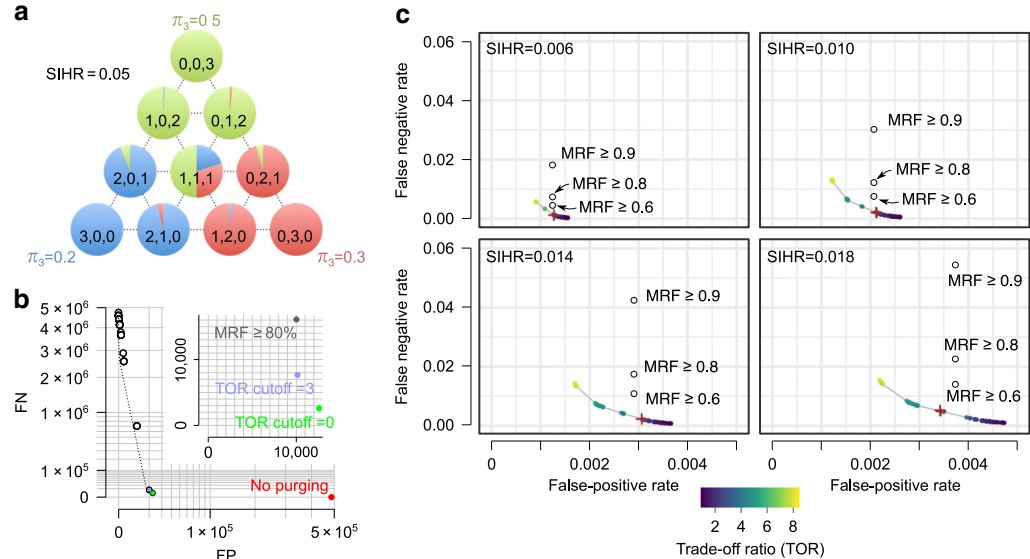

**Fig. 3 Identification and purging of index-hopping artifacts. a** A toy example, depicting the simplex containing all ten possible read count outcomes at PCR amplification level ($r = 3$) and for the case of three samples (see also Supplementary Fig. 8). The pie charts depict the posterior probabilities of the true sample of origin for each outcome. $\pi_3$ represents the proportion of molecules with $r = 3$ that originate from each sample in this toy example. See "Methods" for derivation of the posterior probabilities. **b** Performance on a multiplexed dataset of two libraries with known ground truth (same dataset as Fig. 1c, d). Each dot represents the FN/FP counts, based on the ground truth, at different posterior probability cutoffs. The dashed line represents the model-based estimation of FP/FN. The no purging dot represents the total number of phantom molecules in the uncorrected multiplexed dataset (red dot). The inset shows a zoomed-in view of the graph for posterior probability cutoffs that correspond to trade-off ratio (TOR) cutoffs values of 0 and 3, in addition to a previous heuristic approach[6] that is based on retaining CUGs with a certain fraction of reads assigned to only one sample. **c** Performance on simulated data. The colored dots represent FP and FN proportions for distinguishing true molecules from phantom molecules, after filtering at different posterior probability cutoffs. The color of each dot represents the trade-off ratio (TOR), i.e., the number of real molecules lost for every extra phantom molecule that is correctly purged. The point that corresponds to TOR cutoff = 3 is marked with a plus. The simulations were performed for four evenly spaced values of SIHR (sample index-hopping rate). The open circles show the FN/FP values obtained by the heuristic approach[6] with three choices of the minimum read fraction (MRF) threshold: 0.6, 0.8, and 0.9.

number of mapped reads over total number of molecules (i.e., library size divided by library complexity). Similarly, the samples with the highest RMR also have the highest PPM in both lanes of the Tabula Muris dataset (Supplementary Table 6). This observation suggests that even in multiplexed libraries with small index-hopping rate, low-complexity samples are highly susceptible to large numbers of phantom molecules. Overall, in the datasets that we analyzed, in >25% of samples at least 8% of the UMI counts correspond to phantom molecules (Supplementary Tables 5–6).

**Effects of phantoms on identifying RNA-containing cells.** As mentioned previously, phantom molecules can confound several aspects of the downstream analysis of scRNA-seq data, including identification of RNA-containing cells. To assess the extent to which phantom molecules may affect cell calling, we used the EmptyDrops[9] algorithm to classify cell barcodes in order to determine whether a cell barcode originated from a cell or an empty droplet. For each dataset, we ran EmptyDrops on the unpurged data as well as the purged data (i.e., after removing phantom molecules). The results of this analysis are shown in Supplementary Tables 7–8. For example, we see that for the B1 sample in the HiSeq 4000 dataset[6], before purging the phantom molecules, a total of 1023 cell barcodes are classified as actual cells, whereas rerunning cell calling on purged data produces no more than 16 cells (Supplementary Table 7), suggesting that the majority of the 1023 cells are artifacts of index hopping. These results are consistent with the previous observation that the samples B1 and B2 share many cell barcodes with other samples of the HiSeq 4000 dataset[6], likely due to extensive index hopping.

For the NovaSeq 6000 datasets[8], the effects of contamination are less severe than those seen in the HiSeq 4000 dataset (Supplementary Table 8). Nonetheless, for several samples, the barcodes that are reclassified as empty droplets after purging can make up a substantial fraction of total cells (e.g., samples P7-1 and P7-8). To understand how misclassification of cell barcodes can affect the interpretation of data, we examined the gene expression profiles associated with each cell barcode before purging phantom molecules. As Fig. 4c shows, in two samples of the Tabula Muris dataset[8], we identified a cluster of cells that appear to form a distinct cell type based on their expression profiles; however, the large majority of these cells are reclassified as empty droplets after purging phantom molecules, suggesting that these distinct cell types are artifacts of index hopping.

## Discussion

Our probabilistic framework, as we showed in the "Results" section, provides a principled approach to calculating the index-hopping rate in any given multiplexed droplet-based scRNA-seq dataset. Importantly, it also provides a method for calculating the posterior probability of assignment of each molecule to each sample. Having these posterior probabilities, we can accurately estimate the FP and FN rates for distinguishing true and phantom molecules, without knowing the ground truth about their sample of origin.

As our analyses of several datasets suggest, low-complexity samples in multiplexed datasets often contain a large PPM. We believe that high PPM values might in fact be common in multiplexed datasets that contain both low- and high-complexity samples, since hopping of even a small fraction of reads from

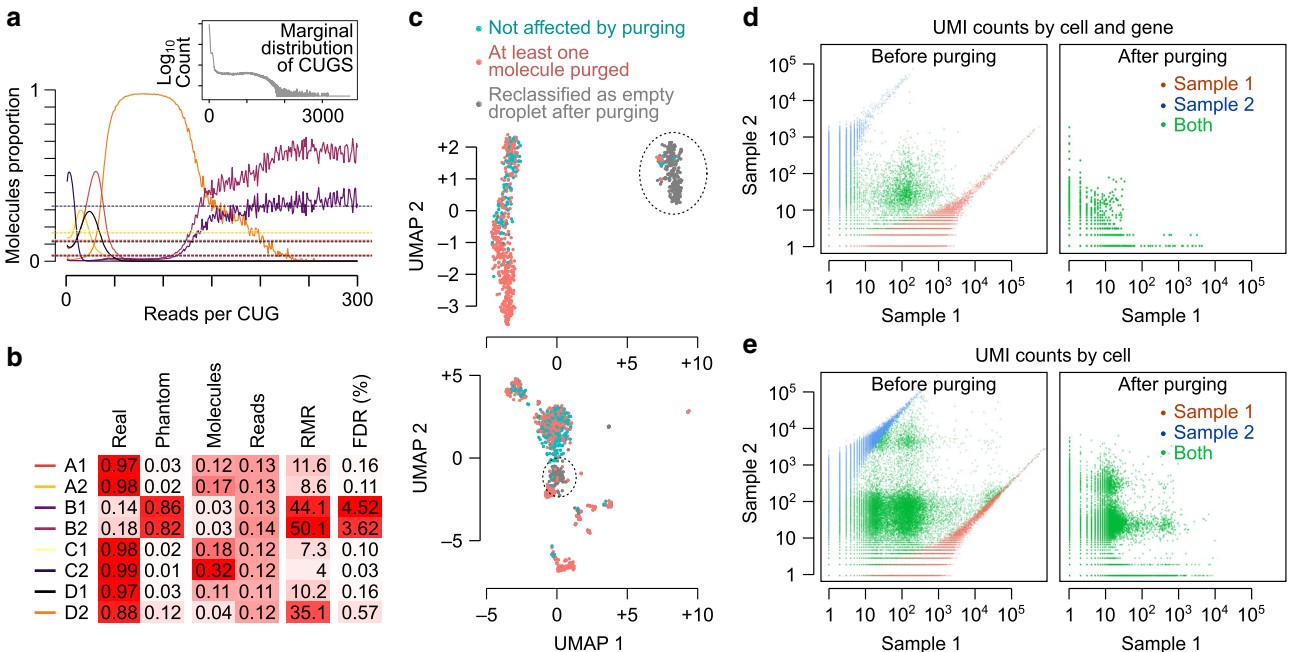

**Fig. 4 Impact of sample index hopping. a** The molecular proportions complexity (MPC) profile of a dataset of eight samples sequenced on HiSeq 4000[7]. Each curve represents one sample, with the y-axis showing the proportion of molecules that belong to that sample conditional on the PCR amplification level (i.e., reads per CUG, r, which is shown on the x-axis). The MPC profile is shown for $r \leq 300$ since there are few CUGs beyond that point to enable accurate estimation of molecular proportions. **b** The proportion of real and phantom molecules in each of the eight samples, the fraction of molecules and mapped reads that belong to each sample, the read-to-molecule ratio (RMR) per sample, and the corresponding FDR statistics (i.e., the within-sample proportion of molecules that we miss-classify as real after purging) for the HiSeq 4000 dataset. See Supplementary Tables 5–6 for additional datasets. **c** Example cases showing the effect of index hopping on cell type identification. The graphs show the UMAP embedding of cells, based on non-purged data, in two samples from the Tabula Muris dataset[8] (top: sample P7-1; bottom: sample P7-8; UMAP embedding based on normalized gene expression profiles). Dotted circles highlight two clusters that are almost entirely made up of cells that are reclassified as empty droplets after purging phantom molecules. **d** The effect of purging on gene expression profiling in the validation dataset with known ground truth. Each dot represents one gene expression measurement in one cell. Dots are colored based on cell-sample assignment in the ground truth, with red and blue representing cell barcodes that are found only in sample 1 or sample 2, respectively. Green represents cell barcodes that are found in both samples (barcode collision). Note that the nonzero sample 1 UMI counts for blue dots and nonzero sample 2 UMI counts for red dots represent phantom molecules. **e** Same as (**d**), but with the counts aggregated per cell (each dots represents one cell). Blue dots with high counts in sample 1 and red dots with high counts in sample 2 represent phantom cells. Green dots represent potential transcriptome mixing.

high- to low-complexity samples can create more phantom molecules than the unique real molecules native to the low-complexity sample. We expect this phenomenon to be widespread, as we observed that in almost all datasets that we analyzed, different samples have widely different MPC profiles (Supplementary Fig. 5).

These phantom molecules can be a substantial confounding factor in multiplexed scRNA-seq experiments. For example, our analysis of the multiplexed dataset with known ground truth indicates that large numbers of genes (Fig. 4d) and cells (Fig. 4e) in each sample are heavily contaminated by phantom molecules. This not only leads to mixing of transcriptomes, but also creates many phantom cells that are entirely made up of molecules hopped from another sample (Supplementary Fig. 7). However, purging of phantom molecules through our model-based approach almost completely remedies the effect of index hopping at the gene and cell level (Fig. 4d, e).

We also found that sample index hopping has a large effect on the accuracy of cell-calling algorithms (i.e., detection of empty droplets from cell-containing droplets). First, a large number of empty droplets appear to be simply artifacts of phantom molecules, and disappear once phantom molecules are purged (Supplementary Tables 7–8). More importantly, a large fraction of droplets that are classified as cells are reclassified as empty droplets once phantom molecules are purged. For example, in one of

the samples from the multiplexed mouse epithelial samples[6], >98% (1007 out of 1023) of the cells are in fact empty droplets that are misclassified due to the abundance of phantom molecules (Supplementary Table 7). This effect can also be seen in other datasets, e.g., in the Tabula Muris dataset[8] where in one of the samples >25% of the called cells are reclassified as empty droplets after purging the phantom molecules (see samples P7-1 in Supplementary Table 8). These artifactual cells can lead to mis-interpretation of data by appearing as novel cell types with a distinct expression profile (Fig. 4c).

Overall, given that sample index hopping appears to be present in almost any multiplexed dataset, and almost always leads to substantial contamination of droplet-based scRNA-seq data at the level of molecule counts, we believe purging of index-hopping artifacts should become a standard procedure. Our probabilistic framework provides a principled, model-based solution to this problem.

Nonetheless, our method has limitations that warrants further improvements (see Supplementary Discussion for a more detailed description of model limitations). First, while the empirical data often fits our model well, at certain ranges and values the data depart from the model's assumptions. For example, as Supplementary Fig. 2 shows, the observed chimeric proportions for the experimental datasets show a downward deviation from the model-predicted mean trend at high PCR amplification levels,

especially for the top 1% of observations that are characterized by noisiness and sparsity. As for the validation dataset, we notice two trends that seem to slightly depart from the model's assumption. First, there is a minor difference between the two samples' proportion of hopped reads, with the SIHR from S1 to S2 being 0.00346 vs. SIHR for S2 to S1 being 0.00302. Second, the ground truth estimates for both samples start out at higher proportion values but stabilize starting at $r = 10$. These trends could be subsampling artifacts of the filtering procedure steps. For example, we retain CUGs that are observed in both the multiplexed and non-multiplexed samples (in order to establish the ground truth), which results in relative depletion of CUGs with low read counts (as they are less likely to be captured in both non-multiplexed and multiplexed data). Alternatively, these trends could reflect an underlying mechanism that the model we have proposed here does not perfectly capture. A model that is governed by several sample-specific hopping rates that are also dependent on the PCR amplification level could provide better accuracy but likely at the cost of intractable computational and mathematical complexity.

Finally, non-negligible probability of CUG collision across samples can be another source of deviation from our model's assumptions when the number of samples is large. As the number of multiplexed samples increases, our first assumption would no longer hold since the probability of observing, in more than one sample, a given CUG label combination becomes non-negligible. That said, in single-cell experiments, multiplexing more than 16 samples on a single lane is not common given that it leads to smaller library sizes and lower genomic coverage. Furthermore, the adoption of longer UMI indexes in more recent droplet-based assays would further reduce the probability of potential collisions, thus rendering this concern less of a problem.

## Methods

A summary of the methods is described here. However, a detailed and extensive statement of the model, its assumptions, motivations, mathematical derivations, and limitations are provided in the Supplementary Notes 1–2, Supplementary Methods, and Supplementary Discussion.

**Modeling sequencing read counts.** We can formally formulate the process of observing index-hopped reads as a two-stage hierarchical sampling process, where we first sample a molecule $s_l$ from a library sample $s_l$ according to the categorical model, then we amplify the molecule by generating $r$ PCR read duplicates according to the multinomial model. We denote the observed vector of read counts with hopped and unhopped reads by $\boldsymbol{y}_l = (y_{l1}, \ldots, y_{ls}, \ldots, y_{lS})$, where $l = \{1, \ldots, m_r\}$ refers to an observed CUG label (i.e., molecule) originating from an unobserved source sample $s_l \in \mathcal{S} := \{1, \ldots, S\}$ and $m_r$ refers to the number of observations with a PCR duplication level $r$. Note that the number of molecules originating from a given sample with a given PCR read duplicates $r$ is determined by the parameter vector $\boldsymbol{\pi}_r$, whereas the number of PCR duplicated reads that end up hopping to other samples is determined by the parameter vector $(\boldsymbol{p}_{s_l})$. As we show in more detail in Supplementary Methods, this hierarchical sampling can be represented as a mixture-of-multinomials model under the assumptions that each CUG represents only one unique real molecule and that sample index hopping is independent of the source or target sample:

$$
\begin{aligned}
\boldsymbol{s}_l &\sim \text{Categorical}(\boldsymbol{\pi}_r) \\
\boldsymbol{y}_l | \boldsymbol{s}_l &\sim \text{Multinomial}(r, \boldsymbol{p}_{s_l}) \\
l &= 1, \ldots, m_r,
\end{aligned}
\tag{1}
$$

where $l$ indexes the $m_r$ observations (molecules) that have PCR duplication level $r$; and the probability vector $\boldsymbol{\pi}_r$ represents the proportion of molecules across the $S$ samples at PCR duplication level $r$. We refer to the set of all molecular proportions $\Pi := \{\boldsymbol{\pi}_r\}_{r=1}^R$ as the MPC profile (see Supplementary Methods). The vector $\boldsymbol{p}_{s_l}$ denotes the probability that a read originating from sample $s_l$ ends up being assigned to each of the $S$ samples, and its elements are equal to $p_h = (1 - p)/(S - 1)$ for samples other than $s_l$, and $p$ for the sample $s_l$ itself, where $p$ is the probability of no index hopping.

**Estimating the sample index-hopping rate (SIHR).** As we show in Supplementary Methods, at any given PCR amplification level $r$, the number of non-chimeric observations, i.e. labels $l$ that are observed in only one sample, follows a binomial

distribution:

$$
z_r = \sum_{l=1}^{m_r} w_l \sim \text{Binomial}\left(m_r, p^r + (S-1) \times \left(\frac{1-p}{S-1}\right)^r\right),
\tag{2}
$$

where $w_l$ is a Bernoulli random variable denoting whether observation $l$ is non-chimeric, and $z_r$ is the sum of counts of non-chimeric observations $l$ at the PCR amplification level $r$. Subsequently, the joint sampling distribution of the non-chimeras at all PCR duplication level values, concatenated as a vector $\boldsymbol{z}$, can be decomposed as follows:

$$
P(\boldsymbol{z}|\theta) = \prod_{r=1}^R \text{Binomial}\left(z_r | m_r, p^r + (S-1) \times \left(\frac{1-p}{S-1}\right)^r\right),
\tag{3}
$$

where $z_r$ is the number of non-chimeric molecules out of the $m_r$ observed molecules at PCR amplification level $r$.

We estimate the value of $p$, the complement of index-hopping rate, by fitting the joint sampling distribution to the observed non-chimera counts, as discussed in Supplementary Methods.

**Reassigning reads and purging phantom molecules.** For each molecule $l$ with observed read counts $\boldsymbol{y}_l$ across $S$ samples, the posterior probability that it originated from sample $s$ can be calculated as:

$$
\mathbb{P}(s|\boldsymbol{y}_l; r, \hat{p}, \hat{\boldsymbol{\pi}}_r) = \frac{\left(\frac{S-1}{1-\hat{p}}\right)^{y_{ls}} \hat{\pi}_{rs}}{\sum_{s=1}^S \left(\frac{S-1}{1-\hat{p}}\right)^{y_{ls}} \hat{\pi}_{rs}},
\tag{4}
$$

where the plug-in estimates $\hat{\pi}_{rs}$ is the proportion of molecules in sample $s$ at PCR duplication level $r$ and $\hat{p}$ is the complement of the SIHR.

We label the sample with the maximum posterior probability as the true sample of origin, the molecule corresponding to that sample as a real molecule, and all others (with nonzero reads) as phantom molecules. Such a procedure achieves the minimum possible number of FN (i.e., real molecules that are incorrectly labeled as phantom), but at the expense of FP (i.e., phantom molecules that are incorrectly labeled as real). To reduce the FP rate, we would need to find a posterior probability cutoff, below which the observations that we labeled as real molecules are also now relabeled as phantom. We select this cutoff based on the trade-off between FP and FN, which are estimated based on the distribution of posterior probability scores. For more details and mathematical derivations see Supplementary Methods.

**Validation data.** Two 10x Genomics scRNA-seq sample libraries were sequenced in two conditions. In the first condition, the samples were multiplexed on the same lane. In the second condition, two sample libraries were sequenced on two separate lanes of HiSeq 4000 (this non-multiplexed condition provides a ground truth for the true sample of origin of each CUG). Cell Ranger 3.0.0 was run with the default options on each of the four samples (two-sample multiplexed on the same lane and the same two samples non-multiplexed). A joined read count table was created. The joined table had 16,547,728 unique CUGs, out of which 9,252,147 CUGs were present in both the multiplexed and non-multiplexed samples. These common CUGs were retained and a column containing the ground truth labels was added to the resulting inner joined data table. Rows corresponding to colliding CUGs were filtered out and the table was saved to file (see the reproducible R markdown notebook validation_hiseq4000_1.nb.html for details).

**Preprocessing of published datasets.** We applied the proposed model on three publicly available 10x Genomics scRNA-seq datasets: (1) a control non-multiplexed dataset in which each sample was sequenced on a separate lane of HiSeq 2500; (2) a multiplexed dataset sequenced on HiSeq 4000; and (3) a multiplexed dataset sequenced on NovaSeq 6000. The HiSeq 2500 and HiSeq 4000 datasets consist of eight libraries of mouse epithelial cells, which have also been used previously[6] for analysis of sample index hopping. The two datasets were downloaded from the authors' host server using the get_data.sh script available on our paper's GitHub repo (for a more detailed description of the data, please refer to the original publication[7]). The third dataset was obtained from the Tabula Muris project's repository. It consisted of 16 libraries (i.e., the P7 libraries) of mouse tissue samples, which were pooled and multiplexed on two lanes of an S2 flowcell in a single NovaSeq 6000 sequencing run[8]. BAM files for the 16 samples were downloaded from the SRA data repository (SRA accession number SRP131661) and converted back to FASTQ files using the 10x Genomics bamtofastq utility. Cell Ranger 3.0.0 was run with the default options on each set of samples multiplexed on the same lane.

**Simulated data.** To further benchmark the performance of the proposed approach and compare with a previous heuristic method[6], we simulated data for ($S = 8$) samples using the MPC profile computed for the mouse epithelial cells HiSeq 4000 data. For computational feasibility, outcomes were simulated for all $r$ values up to a

maximum of $r = 15$ for a range of four SIHR values (0.018, 0.014, 0.010, 0.006). Simulated reads were generated based on our generative probabilistic model.

In order to assess the performance of the approach under a variety of MPC profile instances: we wrote a function that is able to capture the statistical regularities we observe in empirical MPC profiles. We observed that the truncated geometric distribution approximates rather well a sample's molecular proportions especially in the PCR amplification level range between $r = 1$ and $r = 30$. We use the function to extend the simulations to higher PCR amplification level $r = 30$, smaller number of samples ($S = 4$), and for two dissimilar molecular complexity profiles simulated conditions. Simulating and computing the probabilities for all the possible outcomes that make up each Pascal's r-Simplex for a larger range of $r$ is computationally demanding. Furthermore, empirical data tend to be very sparse at this range and the statistical regularities are not easily captured by a simple model such as the truncated geometric distribution. A reproducible analysis notebook showing how the data can be simulated is available at this link https://csglab.github.io/PhantomPurgeR/assets/notebooks/simulation.html.

**Reporting summary**. Further information on research design is available in the Nature Research Reporting Summary linked to this article.

## Data availability
Raw data and read count tables associated with the scRNA-seq validation set are available via GEO (accession GSE149087).

## Code availability
The GitHub project's website, https://csglab.github.io/PhantomPurgeR, contains links to the paper's reproducible analysis notebooks. The associated R package is available at https://github.com/csglab/PhantomPurgeR.

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

## Acknowledgements
The authors thank Alfredo Staffa and James Webber for their comments and useful discussions, and Veena Sangwan for providing the validation samples. This work was supported by the Canadian Institutes of Health Research (CIHR) grant PJT-155966, Brain Canada through the Canada Brain Research Fund with the financial support of Health Canada, and the Alfred P. Sloan Foundation (grant FG-2018-10450) to H.S.N., the Cancer Research UK (CRUK) Grand Challenge funds to L.E.F. and J.R. (project STORMing Cancer), and resource allocations from Compute Canada to H.S.N. H.S.N. holds a CIHR Canada Research Chair. J.R. is supported by funding from the Genome Canada Genome Technology Platform and Canada Foundation for Innovation grants.

## Author contributions
Methodology and conceptualization: R.F. and H.S.N. Mathematical derivation: R.F. with contribution from H.S.N. Formal analysis and code implementation: R.F. Visualization: R.F. and H.S.N. Writing: R.F. and H.S.N. Review and editing: H.D., L.E.F., and J.R. Acquisition of validation data: H.D., L.E.F., and J.R. Directing the study: H.S.N.

## Competing interests
The authors declare no competing interests.
