## [Peer Review File · Nature Communications]

Reviewers' comments:

Reviewer #1 (Remarks to the Author):

Summary:

Farouni et al. present a generative model for estimating the sample index-hopping rate of individual reads and assigning sample of origin for hopped reads in single cell RNA-seq (scRNA-seq) data. Sample assignment is based on posterior probabilities from the model. They generated a ground truth dataset by sequencing two scRNA-seq libraries twice each, once with the two samples multiplexed and sequenced together on the same lane and once with the two samples sequenced in separate lanes. They show that the false negative rate of their model outperforms a previous method by Griffiths et al. (2018). They further showed in simulated data that their method outperforms in sensitivity and specificity. Finally, they analyze the prevalence of phantom molecules in previously published scRNA-seq datasets and find that it can be as high as 85%.

Major Comments:

The method is novel, well-justified, and well-detailed in supplemental material, but the text and figures are somewhat preliminary and do not accurately summarize and convey the key points. Several legends do not describe all basic components of the figures, and some main figures do not serve the intended purpose. Details on specific text errors and figures are below.

One concern is the improvement of their method over Griffiths et al. (2018) ("MRF" method). In the ground truth dataset, the proposed method outperforms MRF in false negative rate, but does about the same or worse in false positive rate. In the simulated data, the proposed method performs better in sensitivity and specificity, but the simulation parameters are quite limited. The authors only simulate from the molecular complexity profile of one dataset and to a PCR amplification level of 15 (while showing that real datasets can get into the thousands). Simulations based on multiple datasets and to a more realistic PCR amplification level would be more convincing.

Minor Comments:

Throughout the paper (most notably in the first sentence of the abstract), the authors claim that their method provides inference of the "true sample of origin" for hopped reads, but this is impossible. The proposed method assigns reads to samples based on the maximum posterior probability of a model, but it can still misassign reads. The true sample of origin is unknowable in most datasets, except for their verification and simulated datasets.

The "as high as 85%" statistic in the abstract is certainly striking, although it seems to be a fairly large outlier (Sample B from the HiSeq 4000 dataset). The rest of the data shows that the fraction of phantom molecules is much lower, e.g. in the Tabula Muris data they report the maximum as 17%. Have the authors followed up at all on these samples? Is there any other supporting evidence that they may be of low quality?

Assumption 2 seems quite reasonable, however it seems to be disproven by the authors' own data (particularly Figs. 1d and 2b). This is acknowledged in the Supplemental Notes' discussion of Assumption 2, but I think it would be best to also mention this in the main Discussion section. In addition to the points made in the Supplemental Notes, why do the two lines in Figure 1d have different lengths (ie. one stops around $r=90$, but the other continues past $r=200$)? This would seem to indicate another way in which the SIHR is not the same across samples.

The possibility that empty droplets may be artifacts of phantom cells is very interesting. While this is mentioned in the abstract and the Discussion section, it is not covered at all in the Results. It

would make sense to either remove it from main text entirely or add a subsection about this to the Results.

The Supplementary Notes provide a very nice breakdown of the proposed method's limitations, but there is none in the main text. A brief summary of these limitations should be added to the Discussion section.

I applaud the authors' extensive work in making their method and results freely available and fully reproducible. I would also like to express interest in trying out their method on additional datasets and (if it is not too much additional work) would encourage the creation of a standalone R package.

Notes on technical terms and abbreviations:

Figure 1b is referenced by the second paragraph of the Introduction and it uses the abbreviation CUG, which isn't defined until the Results section

Phrases such as "molecular proportions complexity profile," "molecules proportion," "complexity profile," and "molecular complexity profile" seem to be used interchangeably throughout the Results and are not defined until the Methods. I found these phrases to be particularly confusing, as they are sometimes treated as a property of a sample (eg. "low-complexity" samples), but π_r is defined by a particular PCR amplification level and contains information on all samples at that level.

y_l is not defined in the Methods. It might also be helpful to also mention x_l , the UMI count.

Having both TOR and TORC seems redundant. Could have just TOR and TOR cutoff. Also, is TORC = 0 the same as "no discarding" (esp. re: Table S2)?

Just before Equation 3, I think "distribution of the chimeras at all PCR..." should be "distribution of the non-chimeras at all PCR..."

Comments on figures:

Figure 1b is very important, but the legend is quite unclear. The left matrix shows read counts (both hopped reads and not), but the legend says "hopped reads (left)." Similarly, the legend says that blue coloration indicates the true sample of origin, but this is misleading, as all CUGs originate from Sample 1, even those which end up with no reads mapping to Sample 1, causing them to appear as *white* zeros.

Figure 1c is not an accurate representation of this data; the areas of the various parts of the Venn diagram are not proportional to the actual numbers.

Figure 1d, as mentioned above, appears to contradict the claim in the text that there is no difference in proportions. The lines look especially different at low reads per CUG, where most of the data is concentrated.

Figure 2a seems somewhat misleading, as it is dominated by the dark red color indicating high probability of chimeric reads, but this is not reflective of what is seen in real datasets. Could consider adding histograms in the margins to show what fraction of reads falls into the area with a high probability of being chimera.

Figure 4a was pretty confusing. "Molecules proportion" (axis label) vs. "complexity profile" (in legend) is unclear. The x-axis only goes up to a PCR amplification level of 300, but the inset histogram shows that there are meaningful numbers of CUGs up to 2000. Some comment should also be made on sample D2, in addition to B1 and B2.

Reviewer #2 (Remarks to the Author):

In this manuscript, the authors describe a statistical method for estimating the rate of index hopping in droplet-based single-cell RNA-sequencing (scRNA-seq) data and removing sequenced molecules which are artifacts due to index hopping. While procedures have been developed for plate-based scRNA-seq data, the introduction of a strategy for droplet-based sequencing methods

is needed. The manuscript is well written and the results are interesting. They designed proper experiments to confirm their statistical assumptions. The methods section and the supplementary information clearly describe their method. We commend them for the inclusion of a github webpage where the analyses to produce the figures for the manuscript and software are already published. We agree with the authors that accounting for potential index hopping in sequencing data should become standard procedure and this work enables this step for droplet-based data. However, we do have a few concerns.

The authors state in the introduction that "identification of cell supopulations (especially rare cell types) as well as genes that are differentially expressed across cell types can be confounded in downstream analyses". Can the authors give any examples of this occurring due to index hopping? Preferably using the data already analyzed in the manuscript, or cite relevant literature.

The authors state in the abstract that a small index hopping probability "counter-intuitively gives rise to a large fraction of "phantom molecules" - as high as 85% in a given sample". They later state in the text that the reason for a fraction this large is due to low sequencing complexity in those samples. We think that the abstract should be revised to clarify the reason for such a high amount of contamination. The text in its current state is somewhat misleading.

Please find below our point-by-point responses to reviewers' comments. Our responses are in blue. In addition, we have provided a merged file that includes the manuscript and associated supplementary materials, in which we have highlighted the major sections of the text that have been modified compared to the previous version.

Here is a brief summary of the changes:

1. New analyses:
 - a. We have included new analyses in response to reviewer #1, including expanded simulations to show that our probabilistic-based method outperforms previous methods in different scenarios, and to simulate a larger range of PCR amplification levels.
 - b. We have performed new analyses in response to reviewers #1 and #2 to explore the effect of misclassification of empty droplets as a result of sample index hopping, showing that this phenomenon creates the appearance of artificial cell types.
2. We have now provided an R package that can be used to measure index-hopping rate and remove phantom molecules from droplet-based scRNA-seq data, in response to reviewer #1.
3. We have edited the text and the figures to address reviewers' concerns about clarity and presentation of the results.
4. We have now merged Supplementary Methods and Supplementary Notes to have a simpler organization, prevent redundancy, and have a better flow in the text. Briefly:
 - a. Section 1 (Background) of the old Supplementary Notes is now moved to Supplementary Methods (section Background).
 - b. Section 2 (Toy data example) was moved from Supplementary Notes to section 2.3 of Supplementary Methods.
 - c. Section 3 (Mathematical Derivations) was moved from Supplementary Notes to section 5 of Supplementary Methods.
 - d. Section 4 (Method's Limitations) is now moved to Supplementary Methods (section 7), with a summary of these limitations added to the end of Discussion in the main text.
 - e. Section 5 (Overview of Computational Workflow) is now embedded in the instructions and vignettes of the R package that we created in response to reviewer #1.
 - f. Section 6 (Data Analysis) was redundant with the Results after moving "The Effects of Phantom Molecules on Identifying RNA-containing Cells" to the main text, and therefore was removed.

Reviewer #1 (Remarks to the Author):

Summary:

Farouni et al. present a generative model for estimating the sample index-hopping rate of individual reads and assigning sample of origin for hopped reads in single cell RNA-seq (scRNA-seq) data. Sample assignment is based on posterior probabilities from the model. They generated a ground truth dataset by sequencing two scRNA-seq libraries twice each, once with the two samples multiplexed and sequenced together on the same lane and once with the two samples sequenced in separate lanes. They show that the

false negative rate of their model outperforms a previous method by Griffiths et al. (2018). They further showed in simulated data that their method outperforms in sensitivity and specificity. Finally, they analyze the prevalence of phantom molecules in previously published scRNA-seq datasets and find that it can be as high as 85%.

Major Comments:

The method is novel, well-justified, and well-detailed in supplemental material, but the text and figures are somewhat preliminary and do not accurately summarize and convey the key points. Several legends do not describe all basic components of the figures, and some main figures do not serve the intended purpose. Details on specific text errors and figures are below.

We thank the reviewer for positive assessment of the method. We have edited the manuscript and performed additional analyses to address the reviewer's concerns, as outlined below.

One concern is the improvement of their method over Griffiths et al. (2018) ("MRF" method). In the ground truth dataset, the proposed method outperforms MRF in false negative rate, but does about the same or worse in false positive rate. In the simulated data, the proposed method performs better in sensitivity and specificity, but the simulation parameters are quite limited. The authors only simulate from the molecular complexity profile of one dataset and to a PCR amplification level of 15 (while showing that real datasets can get into the thousands). Simulations based on multiple datasets and to a more realistic PCR amplification level would be more convincing.

As requested by the reviewer, we have now extended the simulations to higher PCR amplification level and different molecular complexity profiles.

- (a) We have now performed the simulations to a high PCR amplification level of $r=30$. We limited our analysis to $r=30$ because simulating and computing the probabilities for all the possible outcomes that make up each Pascal's r -Simplex for larger range values of r becomes exponentially prohibitive to calculate computationally (see Supplementary Methods, section 2.3 "Toy Data Example" for more details). Furthermore, empirical data tends to be very sparse at this range, and the majority of molecules have PCR amplification level < 30 . In other words, in terms of the PCR amplification level, our simulations now cover $> 99\%$ of the molecules that are found in empirical data.
- (b) We have also extended the simulations to two dissimilar molecular complexity profiles to assess the performance of the approach under different conditions. We first observed that the truncated geometric distribution can accurately approximate the empirical sample molecular proportions, especially in the PCR amplification level range between $r=1$ and $r=30$. We therefore used this distribution to simulate other molecular proportions complexity profiles.

Our new results further support the notion that our method outperforms the MRF approach under a variety of conditions, as shown in the new Supplementary Figure S6. Interestingly, in the new simulations, our probabilistic approach (at TOR cutoff=3) outperforms the MRF method in terms of false positive (FP) rate (Figure S6). We would like to note that, even in the previous simulations (Figure 3c), it is still possible to select a TOR cutoff that would outperform the MRF approach in terms of FP rate. For example,

at SIHR=0.014, any TOR cutoff > 3 would provide a smaller FP than the MRF approach, and many of those cutoffs would also provide a smaller false negative (FN). This is now mentioned in the text (page 6):

“More importantly, the entire range of the FP/FN trade-off curve provides a more optimal choice than the MRF approach, irrespective of the SIHR (Fig. 3c). Furthermore, changing the MRF threshold had little effect on the FP rate, indicating that FP cannot be controlled with this heuristic method.”

A reproducible analysis notebook showing the results and how the data can be simulated is available at this link

<https://csglab.github.io/PhantomPurgeR/assets/notebooks/simulation.html>. The code can be used to perform even more simulations for different molecular proportions complexity profiles.

Minor Comments:

Throughout the paper (most notably in the first sentence of the abstract), the authors claim that their method provides inference of the “true sample of origin” for hopped reads, but this is impossible. The proposed method assigns reads to samples based on the maximum posterior probability of a model, but it can still misassign reads. The true sample of origin is unknowable in most datasets, except for their verification and simulated datasets.

The reviewer is correct in that it is impossible to identify with absolute confidence (i.e. theoretical probability of 1) the true sample of origin of reads, especially for the case of fugue observations as Table S10 shows. What we meant by the phrase “infer” was “probabilistic inference”, i.e. deriving the probability that the true sample of origin of a read is a given sample – we have now corrected the text to convey this point. For example, the abstract now starts by saying *“We introduce a statistical model for estimation of sample index-hopping rate in multiplexed droplet-based single-cell RNA sequencing data and for probabilistic inference of the true sample of origin of the hopped reads”*. We have also made additional modifications in the text to ensure that our text reflects the residual uncertainty in assigning reads to samples.

The “as high as 85%” statistic in the abstract is certainly striking, although it seems to be a fairly large outlier (Sample B from the HiSeq 4000 dataset). The rest of the data shows that the fraction of phantom molecules is much lower, e.g. in the Tabula Muris data they report the maximum as 17%. Have the authors followed up at all on these samples? Is there any other supporting evidence that they may be of low quality?

We have now modified the abstract to more precisely reflect the prevalence of phantom molecules as we observed across different datasets. Specifically, the abstract now says *“in more than 25% of samples, the fraction of phantom molecules exceeds 8%, with this fraction reaching as high as 85% in low-complexity samples”*.

Regarding sample quality, we would like to point out that low quality is only one reason that two samples with a similar number of reads can have very different numbers of molecules across the range of PCR duplication levels. We now mention this issue on page 6:

“Samples may differ in their total number of unique transcripts due to a host of factors, ranging from the presence of varying amounts of RNA that characterize different cell types to accidental errors in library preparation that could cause cells to break up and lose their endogenous mRNA”.

Regarding the quality of samples B1 and B2 from the HiSeq 4000 dataset, Griffiths et al. (2018) also noted irregularities with the B1/2 samples, including a high number of cell barcodes that were shared with other samples, and a rapid drop in the per-cell library size for these samples (reported in their supplementary repo, available at https://raw.githubusercontent.com/MarioniLab/BarcodeSwapping2017/master/bcswap_supp.html). This is now mentioned in the manuscript (page 7):

“... These results are consistent with the previous observation that the samples B1 and B2 share many cell barcodes with other samples of the HiSeq 4000 dataset (Griffiths et al., 2018), likely due to extensive index hopping.”

Assumption 2 seems quite reasonable, however it seems to be disproven by the authors' own data (particularly Figs. 1d and 2b). This is acknowledged in the Supplemental Notes' discussion of Assumption 2, but I think it would be best to also mention this in the main Discussion section. In addition to the points made in the Supplemental Notes, why do the two lines in Figure 1d have different lengths (ie. one stops around $r=90$, but the other continues past $r=200$)? This would seem to indicate another way in which the SIHR is not the same across samples.

We have moved the relevant text from the Supplemental Notes to the Discussion section, to highlight the fact that even though empirical sample index hopping rates from each of the two samples are very close to each other, there is still a difference that warrants development of further methods with sample-specific SIHR parameters (please see page 9).

Regarding the different lengths of the two lines in Figure 1d: Samples tend to have varying ranges of r (PCR amplification levels), perhaps due to differences in library complexity or PCR amplification. In this case, the PCR amplification level for sample S1 reaches >200 , whereas PCR amplification level for molecules from sample S2 stops at ~ 90 . Therefore, we were able to calculate the SIHR for S1-to-S2 hopping at $r>200$, while for S2-to-S1 hopping, the SIHR can be calculated only up to $r=90$. In other words, there are very few molecules in S2 at $r>90$ to begin with, and therefore no hopped reads are detected and SIHR cannot be calculated. This is now indicated in the figure legend: *“Note that the different lengths of the red and blue lines reflect the range of r for which there were enough observations in each sample to calculate the index hopping rate”.* We would like to point out that for the same reason (i.e. sparsity of molecules at high PCR amplification levels), the empirical SIHR estimates become noisy at the right-end of the lines.

The possibility that empty droplets may be artifacts of phantom cells is very interesting. While this is mentioned in the abstract and the Discussion section, it is not covered at all in the Results. It would make sense to either remove it from main text entirely or add a subsection about this to the Results.

We thank the reviewer for this suggestion. The effect of index hopping on classification of droplets was previously discussed in more detail in Supplementary Notes. We have now moved it to the Results, section “The effects of phantom molecules on identifying RNA-containing cells”. We have also performed additional analyses, to show how misclassification of droplets due to phantom molecules can lead to cell subpopulations that are almost entirely made of index hopping artefacts – the new results, based on analysis of two samples in the Tabula Muris dataset, are shown in Fig 4c.

The Supplementary Notes provide a very nice breakdown of the proposed method’s limitations, but there is none in the main text. A brief summary of these limitations should be added to the Discussion section.

We have now added a summary of the limitation section from the Supplementary Notes to the end of the Discussion section, editing the text for clarity and brevity (please see the last two paragraphs of the Discussion, page 9). In the Discussion, we also cite the relevant section of the Supplementary Methods for readers that are interested in more technical aspects of the method limitation.

I applaud the authors’ extensive work in making their method and results freely available and fully reproducible. I would also like to express interest in trying out their method on additional datasets and (if it is not too much additional work) would encourage the creation of a standalone R package.

We thank the reviewer for their interest in our method. We have now created a standalone R package *PhantomPurgeR*, which can be downloaded from here <https://csglab.github.io/PhantomPurgeR/> (please click on “R package” on the top right corner). The website also contains instructions for installation, vignettes that show the workflow, as well as example data.

Notes on technical terms and abbreviations:

Figure 1b is referenced by the second paragraph of the Introduction and it uses the abbreviation CUG, which isn’t defined until the Results section

We added the definition of CUG as unique cell-UMI-gene combination in the legend of Figure 1b.

Phrases such as “molecular proportions complexity profile,” “molecules proportion,” “complexity profile,” and “molecular complexity profile” seem to be used interchangeably throughout the Results and are not defined until the Methods. I found these phrases to be particularly confusing, as they are sometimes treated as a property of a sample (eg. “low-complexity” samples), but π_r is defined by a particular PCR amplification level and contains information on all samples at that level.

We have now changed all mentions of “complexity profile” and “molecular complexity profile” to “molecular proportions complexity profile” (abbreviated to “MPC profile”) throughout the text. In the manuscript, we refer to π_r as the molecular proportions (for each sample) at a given r , while the “molecular proportions complexity profile” is shown by Π_r . Both of these terms were previously defined in the Supplementary Methods; we have now moved the explicit definition from the Supplementary Methods to the Methods

section “Modeling sequencing read counts”. We have also added the definition for MPC in Results section “Comparison with existing phantom purging approaches” (page 6):

“We formulate the library complexity of a set of multiplexed samples as the molecular proportions of the samples conditional on the PCR amplification level, which we term the Molecular Proportions Complexity (MPC) profile.”

y_I is not defined in the Methods. It might also be helpful to also mention x_I, the UMI count.

The summary specification of the model has been rewritten to better match the complete specification given in the Supplementary Methods. Please see Methods section “Modeling sequencing read counts” (page 10).

Having both TOR and TORC seems redundant. Could have just TOR and TOR cutoff. Also, is TORC = 0 the same as “no discarding” (esp. re: Table S2)?

We have now replaced TORC with TOR cutoff throughout the text. We also have clarified in Table S3 legend that TOR cutoff= 0 is the same as “no discarding”. We also added a brief clarification in the third paragraph of the subsection ‘Model-based purging of phantom molecules’ (page 5):

“We evaluated the TOR cutoff approach by contrasting it with two alternative actions: no purging, where we leave the data as it is, and no discarding, where we assign each CUG to the most likely sample (and therefore purge predicted phantom copies of those CUGs) but refrain from further discarding the CUGs whose inferred sample-of-origins have low posterior probabilities (equivalent to a TOR cutoff of zero)”.

Just before Equation 3, I think “distribution of the chimeras at all PCR...” should be “distribution of the non-chimeras at all PCR...”

We thank the reviewer for pointing out this typo – we have now fixed it.

Comments on figures:

Figure 1b is very important, but the legend is quite unclear. The left matrix shows read counts (both hopped reads and not), but the legend says “hopped reads (left).” Similarly, the legend says that blue coloration indicates the true sample of origin, but this is misleading, as all CUGs originate from Sample 1, even those which end up with no reads mapping to Sample 1, causing them to appear as *white* zeros.

We have now re-written the legend for this panel: “A toy example showing a read count matrix (left) and the resulting molecule count matrix (right). In this example, the true sample of origin of all reads is Sample 1. Blue depicts the reads/molecules that are correctly assigned to their true sample of origin, whereas red represents hopped reads/molecules, with the color intensity showing relative counts”. We have also added the definition of chimeric/non-chimeric to the legend.

Figure 1c is not an accurate representation of this data; the areas of the various parts of the Venn diagram are not proportional to the actual numbers.

We have now modified the diagram to make sure the areas are proportional to the actual numbers.

Figure 1d, as mentioned above, appears to contradict the claim in the text that there is no difference in proportions. The lines look especially different at low reads per CUG, where most of the data is concentrated.

We acknowledge in the text that there is a slight difference, but we would like to point out that this difference is still within ~10% of each other. We elaborate on this point when we discuss the method's limitations in Discussion.

Also, we realized that the range of the y-axis (which was from 0.002-0.005) might mislead the reader; we have now modified the range to start from 0, which we believe more accurately depicts the similarity/differences of the two lines.

Figure 2a seems somewhat misleading, as it is dominated by the dark red color indicating high probability of chimeric reads, but this is not reflective of what is seen in real datasets. Could consider adding histograms in the margins to show what fraction of reads falls into the area with a high probability of being chimera.

We agree with the reviewer. We have now added a histogram on top of the graph to show the distribution of r (the x-axis).

Figure 4a was pretty confusing. "Molecules proportion" (axis label) vs. "complexity profile" (in legend) is unclear. The x-axis only goes up to a PCR amplification level of 300, but the inset histogram shows that there are meaningful numbers of CUGs up to 2000. Some comment should also be made on sample D2, in addition to B1 and B2.

The y-axis represents the molecular proportion (for each sample, represented by each of the curves), while the set of curves makes up complexity profile. We apologize that the previous text did not make it clear – we have now modified the legend:

"The molecular proportions complexity (MPC) profile of a dataset of eight samples sequenced on HiSeq 4000. Each curve represents one sample, with the y-axis showing the proportion of molecules that belong to that sample conditional on the PCR amplification level (i.e. reads per CUG, r , which is shown on the x-axis)."

Regarding the range of the x-axis: after $r > 300$, the curves become noisy since they are driven by very few observations that are therefore uninformative. This point is now explicitly indicated in the figure legends. We also realized that we had failed to indicate in the inset histogram that the y-axis is on the log-scale, thus creating the appearance that there is a large number of CUGs up to 2000. We have now fixed the y-axis label on this histogram as well as those in Supplementary Figure S5.

Reviewer #2 (Remarks to the Author):

In this manuscript, the authors describe a statistical method for estimating the rate of index hopping in droplet-based single-cell RNA-sequencing (scRNA-seq) data and

removing sequenced molecules which are artifacts due to index hopping. While procedures have been developed for plate-based scRNA-seq data, the introduction of a strategy for droplet-based sequencing methods is needed. The manuscript is well written and the results are interesting. They designed proper experiments to confirm their statistical assumptions. The methods section and the supplementary information clearly describe their method. We commend them for the inclusion of a github webpage where the analyses to produce the figures for the manuscript and software are already published. We agree with the authors that accounting for potential index hopping in sequencing data should become standard procedure and this work enables this step for droplet-based data. However, we do have a few concerns.

We thank the reviewer for the positive assessment of the novelty and importance of the work. As outlined below, we have now performed additional analyses and modified the text to address the concerns of the reviewer.

The authors state in the introduction that “identification of cell subpopulations (especially rare cell types) as well as genes that are differentially expressed across cell types can be confounded in downstream analyses”. Can the authors give any examples of this occurring due to index hopping? Preferably using the data already analyzed in the manuscript, or cite relevant literature.

To specifically explore the effect of index hopping on droplet-based scRNA-seq data, we have now added a new section to the Results (section “The effects of phantom molecules on identifying RNA-containing cells”), showing that phantom molecules can lead to identification of cell populations that are almost entirely artefact of index hopping. The new results, based on analysis of two samples from the Tabula Muris dataset, are now presented in Fig 4c. A reproducible notebook containing the results has also been uploaded to the paper’s website (https://csglab.github.io/PhantomPurgeR/assets/notebooks/downstream_analysis_umap.html).

Also, as pointed out by the reviewer, previous literature has mentioned the potential effect of index hopping on identification of differentially expressed genes. We had previously cited the relevant literature in Supplementary Notes, section 1.2 “Discovery and Quantification of Index Hopping subsection”. We have now added these citations in the second paragraph of the main text. Namely, the illumina whitepaper (<https://www.illumina.com/content/dam/illumina-marketing/documents/products/whitepapers/index-hopping-white-paper-770-2017-004.pdf>) and Costello et al (2018, doi: 10.1186/s12864-018-4703-0), both of which show that index hopping can cause spurious results in downstream analysis of RNA-seq data (see Figure 5 in illumina paper).

The authors state in the abstract that a small index hopping probability “counter-intuitively gives rise to a large fraction of “phantom molecules” - as high as 85% in a given sample”. They later state in the text that the reason for a fraction this large is due to low sequencing complexity in those samples. We think that the abstract should be revised to clarify the reason for such a high amount of contamination. The text in its current state is somewhat misleading.

We have now revised the abstract as suggested by the reviewer. The abstract now more precisely reflects our observations, and specifically indicates that *“in more than 25% of samples, the fraction of phantom molecules exceeds 8%, with this fraction reaching as high as 85% in low-complexity samples”*.

We would like to point out that it is not necessary to have extreme numbers of phantom molecules to arrive at incorrect conclusions when analyzing scRNA-seq data. For example, as the new Figure 4c shows, in two of the Tabula Muris samples with phantom molecule fractions of 13%-17%, these phantom molecules create artifacts that appear as separate (and seemingly novel) cell clusters, which can have substantial impact on later studies. We think the new analyses that we have added based on the reviewer’s suggestion (Figure 4c) convey this message more clearly.

REVIEWERS' COMMENTS:

Reviewer #1 (Remarks to the Author):

Overall, the authors have done an outstanding job of addressing the concerns raised in the first round of review. I hope they share my opinion that the text and figures are generally much more clear and the case for their method now seems more compelling. The additional work of expanding the simulation study and building a well-documented R package is greatly appreciated. I agree with their belief that “purging of index hopping artifacts should become a standard procedure.”

Major Comment

I appreciate the inclusion of the additional analysis on the effects of phantom molecules for identifying cell types. Since cell type identification is an important part of most scRNA-seq analyses, this is a very important consideration. The new Figure 4c provides an interesting view of potential downstream effects that could be avoided using the authors' work, but it also raises new (perhaps tangential) questions.

I do not understand the justification for running UMAP on individual samples rather than the complete dataset (which is how UMAP is generally used and also how PhantomPurgeR seems to be intended). I suspect that it may be more useful to show UMAP plots run on all of the samples before and after purging, to demonstrate how removing phantom reads and empty droplets improves the resolution of different cell types. This could potentially provide a very clear example of how the authors' method would improve any scRNA-seq analysis pipeline.

Minor Comments

Tables S7 and S8 are confusing and warrant additional attention since they comprise the bulk of the results in the final Results section. In the main text it is stated that for Sample B1 in the HiSeq 4000 dataset, 1,023 cell barcodes are classified as actual cells, whereas after rerunning cell-calling on the purged data there are no more than 16 cells. However there is no “1,023” number visible in the table. To be more consistent, there should be a clear indication in the table of which columns represent these “before” and “after” numbers in the main text and table caption.

In addition, tables S7 and S8 contain a few inconsistently punctuated numbers (“195614”, “19,69”, “10,89”).

The first sentence of the abstract assumes the reader already has some knowledge of sample index hopping. While this is not unreasonable, the authors may want to consider adding an introductory sentence explaining the phenomenon. (Personally, I had never heard of it before reading this paper, but I am now convinced that it is an issue that deserves attention).

The paper uses two different versions of the word artifact/artefact.

Reviewer #2 (Remarks to the Author):

The authors have answered all our questions and I have no further concerns.

Please find below our point-by-point responses to reviewers' comments. Our responses are in blue.

Reviewer #1 (Remarks to the Author):

Overall, the authors have done an outstanding job of addressing the concerns raised in the first round of review. I hope they share my opinion that the text and figures are generally much more clear and the case for their method now seems more compelling. The additional work of expanding the simulation study and building a well-documented R package is greatly appreciated. I agree with their belief that "purging of index hopping artifacts should become a standard procedure."

We thank the reviewers for the constructive comments, and completely agree that by addressing the referee comments the text and figures are considerably improved now.

Major Comment

I appreciate the inclusion of the additional analysis on the effects of phantom molecules for identifying cell types. Since cell type identification is an important part of most scRNA-seq analyses, this is a very important consideration. The new Figure 4c provides an interesting view of potential downstream effects that could be avoided using the authors' work, but it also raises new (perhaps tangential) questions.

I do not understand the justification for running UMAP on individual samples rather than the complete dataset (which is how UMAP is generally used and also how PhantomPurgeR seems to be intended). I suspect that it may be more useful to show UMAP plots run on all of the samples before and after purging, to demonstrate how removing phantom reads and empty droplets improves the resolution of different cell types. This could potentially provide a very clear example of how the authors' method would improve any scRNA-seq analysis pipeline.

We ran UMAP on individual samples to more faithfully capture real-world situations that may arise. For example: (a) samples from different research groups might be multiplexed on the same lane, and then the demultiplexed results are sent back to each group separately (this is in fact how we first came across the problem of sample index hopping); (b) samples from unrelated projects might be multiplexed, and then each sample is analyzed independently; (c) different samples might represent fundamentally different units (e.g. different organs), and therefore phantom cells that are hopped across samples might confound the interpretation of the results. Overall, these scenarios imply that it is important to understand how phantom cells affect each of the samples in a pooled library, which is why Figure 4c shows the per-sample effect.

Minor Comments

Tables S7 and S8 are confusing and warrant additional attention since they comprise the bulk of the results in the final Results section. In the main text it is stated that for Sample B1 in the HiSeq 4000 dataset, 1,023 cell barcodes are classified as actual cells, whereas after rerunning cell-calling on the purged data there are no more than 16 cells. However there is no “1,023” number visible in the table. To be more consistent, there should be a clear indication in the table of which columns represent these “before” and “after” numbers in the main text and table caption.

We have added two columns in the tables showing the number of cells before and after purging as suggested by the reviewer.

In addition, tables S7 and S8 contain a few inconsistently punctuated numbers (“195614”, “19,69”, “10,89”).

We have fixed the punctuation errors.

The first sentence of the abstract assumes the reader already has some knowledge of sample index hopping. While this is not unreasonable, the authors may want to consider adding an introductory sentence explaining the phenomenon. (Personally, I had never heard of it before reading this paper, but I am now convinced that it is an issue that deserves attention).

We have added an introductory sentence to the abstract.

The paper uses two different versions of the word artifact/artefact.

We have now used “artifact” throughout the manuscript.

Reviewer #2 (Remarks to the Author):

The authors have answered all our questions and I have no further concerns.